# Kinematic characteristics of the tennis serve from the ad and deuce court service positions in elite junior players

**Janina Fett**[ORCID]*, **Nils Oberschelp, Jo-Lâm Vuong, Thimo Wiewelhove, Alexander Ferrauti**

Department of Training & Exercise Science, Faculty of Sport Science, Ruhr University Bochum, Bochum, Germany

* janina.fett@rub.de

**Data Availability Statement:** All relevant data are within the manuscript and its Supporting Information files.

## Abstract

### Purpose

According to the official rules of the International Tennis Federation, players have to serve alternately from two different positions: the deuce (right, D) and the ad court (left, AD) side. This study aimed to compare body and ball kinematics of flat serves from both service sides.

### Methods

In a controlled, semi-court laboratory setting, 14 elite male junior players served eight flat first serves to a target field directed to the receiver's body from both service positions in a matched and counterbalanced order. An 8-camera-Vicon-System was used to capture the 3D-landmark trajectories.

### Results

The mean service velocity was found to be similar on both sides (D: 151.4 ± 19.8 vs. AD: 150.5 ± 19.4 km/h), while multiple characteristics of the serve and ball kinematics differed significantly ($p < .05$). At starting, the front-foot angle relative to the baseline (D: 39.7±17.6˚ vs. AD: 31.1±17.4˚) and lateral distance between the feet (D: 16.3 ± 12.9 cm vs. AD: 26.2 ± 11.9 cm) were significantly different. During the service, upper torso range of motion from maximum clockwise rotation until impact was significantly greater on the deuce court (D: 130.5 ± 19.8˚ vs. AD: 126.7 ± 21.1˚). This was especially pronounced in foot-back technique players. Further, differences in the lateral ball impact location (D: 30.0 ± 24.1 cm vs. AD: 10.3 ± 23.3 cm) were observed.

### Conclusions

Changing the service side affects the serve and ball kinematics in elite junior tennis players. Our results underline biomechanical differences regarding the starting position (feet and upper torso) as well as the movement and ball kinematics which could be relevant for skill acquisition, injury prevention and performance enhancement.

**Funding:** The present study was funded by the German Federal Institute of Sport Science (http://www.bisp.de). The Grant number was AZ-072017/16. Funding was received by AF. We further acknowledge support by the Open Access Publication Funds of the Ruhr-Universität Bochum. The funders had no role in study design, data collection and analysis, decision to publish, or preparation of the manuscript.

**Competing interests:** The authors have declared that no competing interests exist.

## Introduction

In tennis, all points start with the serve, which has become the most important stroke and a key factor of game success [1–3]. This stroke allows players to gain points with very short rallies and percentage of points won after the first serve correspond to around 72–81% [4, 5]. Additionally, a successful first serve has evolved into a powerful tool to achieve direct points or to take instant initiative within a rally [6]. Nevertheless, it is also the most difficult stroke to master due to the complex combination and coordination of limb and joint movements required to summate and transfer forces from the ground up into the racquet head, reported as the kinetic chain [7]. Therefore, an appropriate skill acquisition (i.e., movement execution, technical skill, coordination of kinetic chain) is needed from beginners to elite junior players to ensure the best possible outcomes regarding a powerful and efficient serve.

According to the official rules of the International Tennis Federation (ITF), players must serve alternately from the two different positions, the deuce (right) and the ad court (left) side, into the diagonally opposite service boxes [8]. The percentage of serves performed per match from either side is almost identical (47.6% and 52.4% from AD and D, respectively) since only games ending after a 40:15 or 15:40 result in a higher number of serves from the deuce court side. Depending on the tactical strategy, different locations can be targeted: out wide, to the receiver's body, and to the T (i.e., near the centre service line) [9].

Interestingly, from a biomechanical point of view, only one service model is mediated [7], not taking into account the service side. Usually, the serve is characterized by a corkscrew motion: after the ball toss, the serving arm moves behind the body, and the vertebral column is laterally flexed and hyperextended with a fully loaded lower body position [10]. Acceleration of the serving arm and racquet before ball impact is accompanied by a fast counter-rotation of the lumbar spine—from hyperextension to flexion, and from right twist to left twist [7, 10, 11]. Under the current recommendations for the starting position, players are instructed to put their front foot towards the right net post, and their rear foot parallel to the baseline (for a right-handed player). In an attempt to achieve stability, a common advice is to place the toes of the rear foot in line with the heel of the front foot [12, 13].

Due to its great importance, the serve has received significant biomechanical interest. A large number of investigations have examined the kinematic characteristics of lower and upper limb and trunk joint motion [1, 3, 10, 14–18], racquet and ball kinematics [2, 19–22], kinematics in relation to performance level, gender, age, and injury [10, 23–26] and effect of serve type (flat, kick, slice) [27, 28]. A small body of literature has focused on differences in serving locations [20, 22], but no study has examined serve kinematics according to the different service sides: the ad versus the deuce service side. It can be hypothesized that changing the service side can lead to different body and ball kinematics. To the best of our knowledge, this is the first study to examine serve kinematics from both the deuce and ad court sides. Thus, this study aimed to compare the body and ball kinematics of flat serves from the deuce and ad court side, targeting the centre of the service box (i.e., serving to the receiver's body).

## Materials and methods

### Subjects

Fourteen male elite junior squad players of the German Tennis Federation (age: 14.6 ± 1.8 years, age at peak height velocity (APHV): 14.0 ± 0.7 years, maturity offset (MO): 0.6 ± 1.9, weight: 61.4 ± 16.3 kg, height: 176.0 ± 15.9 cm) participated in this study. Twelve players were right-handed and two left-handed. Considering the foot technique, there were six foot-up and eight foot-back technique players. Players were ranked on the national youth ranking list and

had a weekly training volume (without competition) of 10.0 ± 2.6 hours on tennis specific training (i.e., technical and tactical skills). Participants were excluded if they had musculoskeletal injury (upper extremity surgery, shoulder, back, knee, ankle pain) within the past 12 months, conduct any sport-related rehabilitation during the 12 month prior the study or had any other kind of pain during the service execution.

This study was approved by the ethics committee of the Faculty of Sport Science of the Ruhr University Bochum (EKS-24072017), and all procedures conformed to the recommendations and guidelines of the Declaration of Helsinki. The players and parents were fully informed of all experimental procedures, and both players and parents provided written informed consent before participation.

## Experimental design

**Procedures.** The investigation was conducted in a specific semi-court indoor tennis laboratory on a hard-court surface (Rebound Ace). Players had to serve against an absorption wall with a target field (direction to the receiver's body from both sides). The position and size of the target field on the absorption wall were calculated such that it would be a valid serve on a full-size tennis court directed to the receiver's body. For this calculation, triangulation was used to transfer the specific dimensions to the laboratory setting. A schematic screen is available as S1 Fig.

Each player was fitted with 86 retro-reflective markers placed on anatomical landmarks using double-sided tape according to the UWA full-body marker set [26], which is further characterized by a cluster method that saw three markers attached to each segment [16]. Additionally, five markers were fixed to the players' racquets to create coordinate systems therein [26], and retro-reflective tape was placed on the ball in order to determine ball data [29]. To limit movement of the markers from their anatomical landmarks, all players wore tight shorts only. Familiarization with the testing surrounding and the landmark set has been implemented prior to the experiment.

All players completed a standardized warm-up protocol prior to testing. The standardized warm-up consisted of general movement preparation exercises, specific activation and mobilization with elastic tubes, and a set of submaximal serves with increasing velocities (total: 32; 16 at 50–70%, eight at 70–80%, and eight at 90–100% in accordance to their subjective feeling). For the testing protocol each player performed two sets of eight maximum "first flat" serves to the target field (direction: receiver's body; size: $67 \times 30$ cm). The service execution was performed from both service sides (deuce and ad court) in a matched and counterbalanced order. After each set, there was a two minute rest period, between each serve there was a rest of 30s to prepare for the next service. Participants were instructed to serve the ball exactly as they would during a match with maximum power into the predefined target zone. To maximize ecological validity, players used their own racquets to feel as comfortable as possible during their serves [26].

The three fastest serves that landed in the target area were analysed [30]. Four players failed to make three valid attempts within eight serves. Because of that four players had to perform in total (both sides) three to four further serves to get three serves in on each side. Serve velocity was measured using a radar gun (Stalker Professional Sports Radar; Radar Sales, Plymouth, MN). The radar was located 2 m behind the server. It was aligned with the approximate height of ball contact (~ 3 m) and was aimed down the target zone. Peak velocity of each stroke was recorded.

**Kinematic analysis.** An 8-camera Vicon Vantage V5-System operating at 300 Hz was used to capture the three-dimensional (3D) landmark trajectories to reconstruct the service

motion. After capturing all static and dynamic trials, the trajectories were reconstructed with Vicon Nexus software (Nexus, Vicon, Oxford, UK). The experimental data analysis was conducted on the basis of prior kinematic studies carried out by a research group of the University of Western Australia [17, 19, 21–23, 26]. According to the methods prescribed in their previous work [17, 22, 26], gaps were interpolated using a cubic spline; data were filtered using a Woltring filter and subsequently modelled with a customized version of the University of Western Australia model to calculate relevant anatomical, racquet, and ball data [16, 17, 19, 21–23, 26]. Ball data were held relative to the front foot. Therefore, the origin of the global coordinate system was translated to the position of the first metatarsal marker (prior to the initiation of each participant's backswing) [22]. Negative x pointed lateral to the right along the baseline, negative y pointed towards to the net, and positive z pointed vertically upwards. The Euler Z-X-Y sequence was used to describe joint rotations, except at the shoulder where Y-X-Y decomposition was applied to estimate shoulder joint motion, as recommended by the International Society of Biomechanics [16, 17, 19, 31]. The kinematic data of left-handed players were inverted where appropriate [17, 26].

**Kinematic variables.** The service action was divided into different parts to analyse kinematic variables of interest: (1) starting position, (2) preparation phase, (3) propulsion, and (4) impact. Starting position was defined as when the player begins to initiate the ball toss. Time phase until peak shoulder external rotation was defined as the *preparation phase;* the phase from this key point to impact was defined as *propulsion* [17]. All variables of interest are shown in S1 Appendix.

During the starting position, the position of the feet and upper torso axis was measured relative to the baseline (x-axis of the global coordinate system). Upper torso rotation was calculated by a vector joining the two shoulder joint centres, which was expressed relative to the global x-axis [21]. From an overhead view, forward rotation (+) was counter-clockwise, and backward rotation (-) was clockwise for right-handed players. Foot axis was calculated by a vector joining the middle of the forefoot and the heel. In addition, the lateral distance between both toes (first metatarsal) was measured.

During the preparation phase, the analysed parameters were knee flexion, trunk extension, trunk tilt, external shoulder rotation, and elbow flexion. In addition, maximum upper torso position relative to the baseline and counter- upper torso rotation range of motion (ROM) was measured. Counter-upper torso ROM was defined as the ROM between upper torso starting position (relative to baseline) and maximum upper torso position (relative to baseline).

The peak angular velocities during the propulsion phase were focused on the knee extension, trunk flexion, trunk tilt, elbow extension, as well as shoulder internal rotation.

At the time of impact, knee extension, trunk extension, trunk tilt, upper torso position relative to baseline, upper torso ROM, shoulder abduction, and elbow extension were analysed. Upper torso ROM was defined as the maximum backward rotation (during preparation) to forward rotation until impact. The ball kinematics at impact were calculated relative to the first metatarsal of the front foot at the instant of starting position [19, 22].

S2 Appendix shows reliability data for all analysed variables. Overall, 'good' and 'excellent' reliability was found for nearly each variable (ICCs ranging from 0.76 to 0.99). Front knee extension velocity and shoulder internal rotation velocity during propulsion as well as front and back knee flexion at impact showed a 'moderate' reliability (ICCs ranging from 0.58 to 0.73) [32].

**Anthropometric measurements.** Participants body height and sitting height was measured to the nearest mm with a fixed stadiometer (Holtain Ltd., Crosswell, UK). Further, a purpose-built table was used for measuring sitting height. Body mass was recorded to the nearest 0.1 kg with a bioelectrical-impedance scale InBody770 (InBodyCo., Ltd., Gangnam-gu, Seoul,

Korea). Status of maturity was calculated according to the maturity offset method as previously described [33, 34].

## Statistical analysis

All data are presented as mean values and standard deviations (± SD). Paired sample $t$-tests were used to determine differences in kinematics between both sides, the deuce and ad court side. Additional, in case of non-normality the Wilcoxon-test was used. Further, the standardized difference or effect size (ES) of changes in each parameter between the two groups were calculated using the pooled standard deviation. Threshold values for Cohen's d ES statistics were < 0.2 (small), 0.5 (moderate), and > 0.8 (large) [35]. For subgroup analysis of the total upper torso ROM (foot-up and foot-back players), a repeated-measures ANOVA was used. Intra-session reliability of all variable measures was calculated. Intraclass correlation coefficient (ICC 3,1) and standard error of measurement (SEM) (90% confidence limits) were assessed by the three testing trials and pooled between both service sides using the spreadsheets for analysis of validity and reliability of Hopkins [36]. Statistical significance was set at $p \leq .05$. Data analyses were performed using the free statistical software JASP (version 0.11.1) and Microsoft Office 365 MSO (version 16.0.13029.20232).

## Results

### Serve kinematics

Mean values (± SD) of serve and ball kinematics for both conditions (deuce and ad court) are presented in Table 1. Comparing body kinematics on both service sides, there were significant differences in foot position as well as upper torso position at starting position. The front-foot angle (relative to baseline) was higher on the deuce court compared to the ad court side (deuce: 39.7 ± 17.6˚ vs. ad: 31.1 ± 17.4˚; ES 0.49). The lateral distance between both feet was less on the deuce court (deuce: 16.3 ± 12.9 cm vs. ad: 26.2 ± 11.9 cm; ES -0.80) (Fig 1). Upper torso position at starting position was higher on the ad court (deuce: -60.9 ± 15.7˚ vs. ad: -69.6 ± 15.0˚; ES -0.57).

During preparation, higher max. upper torso position (clockwise) was found on ad court side (deuce: -105.6 ± 9.5˚ vs. ad: -120.5 ± 10.3˚; ES -1.50). Further, counter-upper torso ROM was higher on ad court side (deuce: 44.7 ± 15.3˚ vs. ad: 50.9 ± 16.1˚; ES -0.39).

At impact, there was a higher knee flexion in the front leg on the deuce court side (deuce: 29.1 ± 10.3˚ vs. ad: 26.2 ± 10.4˚; ES 0.28). Upper torso ROM from maximum clockwise rotation until impact was significantly greater on the deuce court side (deuce: 130.5 ± 19.8˚ vs. ad: 126.7 ± 21.1˚; ES 0.18). Additionally, the alignment of the upper torso at impact was to be rotated significantly further forward on the deuce court side relative to baseline (deuce: 24.9 ± 16.5˚ vs. ad: 6.2 ± 16.6˚; ES 1.13).

Additional subgroup analysis revealed descriptively slightly higher side differences in the upper torso ROM for foot-back players (n = 8, diff = 4.8˚ ± 4.8˚, $p = .026$) compared to foot-up players (n = 6, diff = 2.4 ± 4.2˚, $p = .215$). However, the side × subgroup interaction was unclear ($p = .360$) (Table 2 and Fig 2).

### Ball kinematics

Mean service velocity was similar from both sides (deuce: 151.4 ± 19.8 km/h vs. ad: 150.5 ± 19.4 km/h; ES 0.04). Lateral ball impact location (X) was more leftwards on the deuce court (deuce: 30.0 ± 24.1 cm vs. ad: 10.3 ± 23.3 cm; ES 0.83) from a back-view position (Table 1).

**Table 1. Peak and terminal kinematic variables of interest for the ad and deuce service side.**

| | | Advantage court | | | Deuce court | | | |
|---|---|---|---|---|---|---|---|---|
| | Unit | Mean | | SD | Mean | | SD | *p* | ES |
| **Starting position** | | | | | | | | | |
| Front foot position to baseline | [deg] | 31.1 | ± | 17.4 | 39.7 | ± | 17.6 | .000 | 0.49 |
| Back foot position to baseline | [deg] | 12.0 | ± | 9.8 | 12.0 | ± | 6.6 | .988 | -0.01 |
| Lateral feet distance § | [cm] | 26.2 | ± | 11.9 | 16.3 | ± | 12.9 | .000 | -0.80 |
| Upper torso position to baseline | [deg] | -69.6 | ± | 14.6 | -60.9 | ± | 15.7 | .002 | -0.57 |
| **Preparation** | | | | | | | | | |
| Front knee flexion | [deg] | 69.5 | ± | 15.0 | 71.6 | ± | 15.7 | .025 | 0.14 |
| Back knee flexion | [deg] | 75.0 | ± | 10.6 | 77.6 | ± | 9.6 | .067 | 0.26 |
| Trunk extension | [deg] | -44.0 | ± | 10.6 | -44.2 | ± | 10.3 | .548 | -0.02 |
| Trunk tilt | [deg] | 19.2 | ± | 6.5 | 19.4 | ± | 5.8 | .649 | 0.03 |
| Max. upper torso position | [deg] | -120.5 | ± | 10.3 | -105.6 | ± | 9.5 | .000 | -1.50 |
| Counter-upper torso rotation [#] | [deg] | 50.9 | ± | 16.1 | 44.7 | ± | 15.3 | .020 | -0.39 |
| Shoulder external rotation | [deg] | 138.1 | ± | 11.4 | 136.7 | ± | 10.6 | .125 | 0.13 |
| Elbow flexion | [deg] | 132.2 | ± | 10.4 | 132.7 | ± | 9.8 | .553 | 0.05 |
| **Propulsion** | | | | | | | | | |
| Front knee extension $\omega$ | [deg/s] | 0443.6 | ± | 108.0 | 0447.2 | ± | 99.1 | .756 | -0.03 |
| Back knee extension $\omega$ | [deg/s] | 0540.4 | ± | 80.4 | 0517.9 | ± | 101.9 | .066 | 0.25 |
| Trunk flexion $\omega$ | [deg/s] | 0506.7 | ± | 69.0 | 0493.2 | ± | 71.2 | .111 | -0.19 |
| Trunk tilt $\omega$ | [deg/s] | 0421.0 | ± | 99.6 | 0424.0 | ± | 96.5 | .696 | -0.03 |
| Shoulder internal rotation $\omega$ | [deg/s] | 1970.9 | ± | 275.9 | 2028.7 | ± | 331.7 | .203 | -0.19 |
| Elbow extension $\omega$ | [deg/s] | 1546.5 | ± | 303.1 | 1563.6 | ± | 327.0 | .584 | -0.05 |
| Wrist flexion $\omega$ | [deg/s] | 1095.4 | ± | 339.9 | 1070.9 | ± | 299.5 | .439 | -0.08 |
| **Impact** | | | | | | | | | |
| Front knee flexion | [deg] | 26.2 | ± | 10.4 | 29.1 | ± | 10.3 | .003 | 0.28 |
| Back knee flexion $1 | [deg] | 5.6 | ± | 8.1 | 6.7 | ± | 8.2 | .474 | 0.13 |
| Trunk extension | [deg] | -8.0 | ± | 9.6 | -7.6 | ± | 9.5 | .327 | 0.05 |
| Trunk tilt | [deg] | -27.6 | ± | 4.4 | -27.2 | ± | 4.1 | .482 | 0.07 |
| Upper torso position to baseline | [deg] | 6.2 | ± | 16.6 | 24.9 | ± | 16.5 | .000 | 1.13 |
| Upper torso rotation (ROM) | [deg] | 126.7 | ± | 21.1 | 130.5 | ± | 19.8 | .008 | 0.18 |
| Shoulder abduction | [deg] | 114.5 | ± | 6.4 | 114.0 | ± | 6.4 | .427 | -0.07 |
| Elbow flexion | [deg] | 18.0 | ± | 8.5 | 18.0 | ± | 7.8 | .998 | 0.00 |
| Wrist extension $2 | [deg] | -20.5 | ± | 6.9 | -20.3 | ± | 6.2 | .796 | 0.03 |
| **Ball kinematics** | | | | | | | | | |
| Ball velocity | [km/h] | 150.5 ± 19.4 | ± | 19.4 | 151.4 | ± | 19.8 | .505 | 0.04 |
| Ball impact location X (lateral) | [cm] | 10.3 | ± | 23.3 | 30.0 | ± | 24.1 | .000 | 0.83 |
| Ball impact location Y (forward) | [cm] | -49.9 | ± | 17.7 | -51.6 | ± | 20.9 | .552 | -0.09 |
| Ball impact location Z (upward) | [cm] | 260.0 ± 23.1 | ± | 23.1 | 258.0 | ± | 22.7 | .086 | -0.08 |

$1 non-normality distribution: Wilcoxon Test: *p* = .194; $2 non-normality distribution: Wilcoxon Test: *p* = .241

§ first metatarsal of front foot to first metatarsal of rear foot; # range of motion between upper torso starting position and maximum upper torso position during preparation; $\omega$ maximum angular velocity; ES effect size

## Discussion

This is the first study to examine the effects of the deuce and the ad court side on body and ball kinematics during flat serves in elite junior tennis players. Obtaining knowledge about the respective serve characteristics is quite important since, according to the rules of the ITF, the

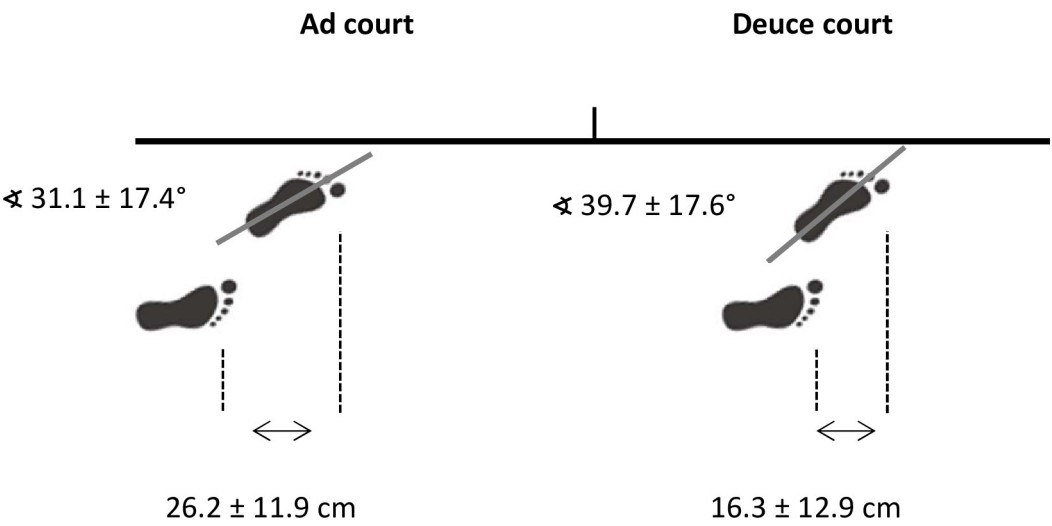

**Fig 1. Mean foot position on ad and deuce court side immediately before ball toss.**

distribution of serves performed from both sides is similar for almost every game score (except for a game win after 40:15 or 15:40). Our data demonstrate various significant differences regarding the starting position, as well as serve kinematics, when changing the service side. Specifically, these differences involve the foot starting position (front-foot angle to baseline), the upper torso position in relation to baseline, the total upper torso ROM until impact, the knee flexion of the front leg at impact and the lateral ball impact point. However, no differences were found for service speed (Table 1). These deviations are surprising, since in most coaches' textbooks only one service technique is taught [7, 12, 13].

Regarding the basic foot position prior to the service movement, the front-foot and back-foot axis seem largely similar on the deuce and ad court side when comparing foot angle in relation to the baseline (Table 1 and Fig 1). A similar service basic position related to the baseline is recommended in the tennis literature (i. e. the front foot is aligned towards the right net post, and the rear foot is placed parallel to the baseline) [12, 13], with acceptable inter-individual variations [7] and is widely accepted by professional players and coaches. Interestingly, during stroke preparation, propulsion and ball impact, we found a significantly higher upper torso ROM until ball impact, with higher knee flexion of the front leg on the deuce court side (Table 1 and Figs 3 and 4). These changes were accompanied by a more leftwards ball impact point (Fig 4).

In general, our kinematic data are comparable to previous findings [2, 18, 21, 22]. Nevertheless, the small, but significant, kinematic differences found between both service sides need to be discussed in greater depth. As this is the first study to focus on this question, a comparison with other studies is limited. We therefore assume that the higher upper torso ROM on the

**Table 2. Upper torso ROM in foot-up and foot-back players.**

| Upper torso rotation (ROM) | | Advantage court | | Deuce court | | Difference | | | Side x group |
|---|---|---|---|---|---|---|---|---|---|
| | Unit | Mean | SD | Mean | SD | Mean | SD | p | p |
| Foot up (n = 6) | [deg] | 129.5 ± 23.1 ± | 24.1 | 131.9 ± | 23.2 | 2.4 ± | 4.2 | .215 | .360 |
| Foot back (n = 8) | [deg] | 124.7 ± | 20.0 | 129.5 ± | 18.4 | 4.8 ± | 4.8 | .026 | |

SD: standard deviation; ROM: range of motion

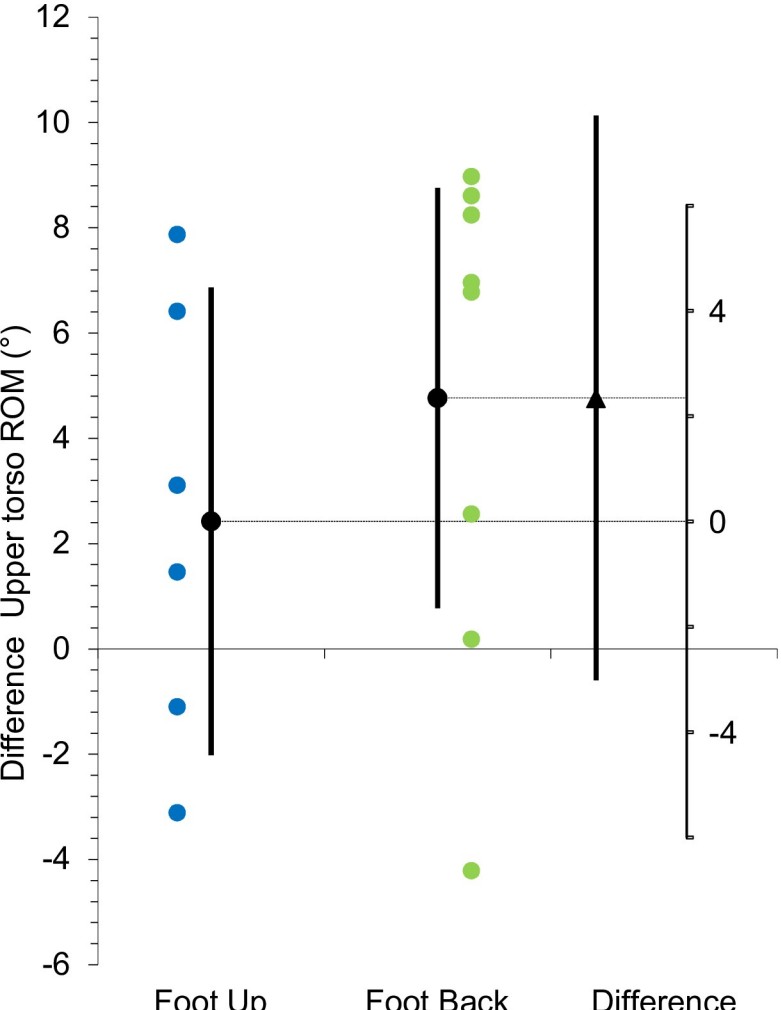

**Fig 2. Subgroup analysis of side differences in upper torso range of motion between foot-up and foot-back players (95% CI).**

deuce court side compared to the ad court side, as shown in our study (Table 1 and Figs 3 and 4), is mainly affected by the relationship between the basic foot and upper torso position and the respective target point in the diagonally opposite service box. Obviously, despite a (only) marginal adjustment of the starting position to a more open stance on the deuce court (Fig 1), this trend does not meet the complete requirements for an identical shaping of the service movement on both sides (i.e. similar upper torso ROM).

Regarding the underlying reasons for these surprising results, it has to be considered that the service is a highly asymmetric task from a perceptual point of view. The spatial–visual orientation differs significantly between the two service sides. During preparation, the visual perception is decreased on the deuce court side in terms of the target field during the early preparation phase which may have an effect on the player´s behaviour and the biomechanical outcome [37]. It further can be speculated that, for most players and coaches, the baseline seems to be the first point of reference for the foot position during serve preparation. Historically, various top players have even taken a parallel foot position entirely behind the baseline on both service sides (e. g. John McEnroe, USA). From a biomechanical point of view, and to

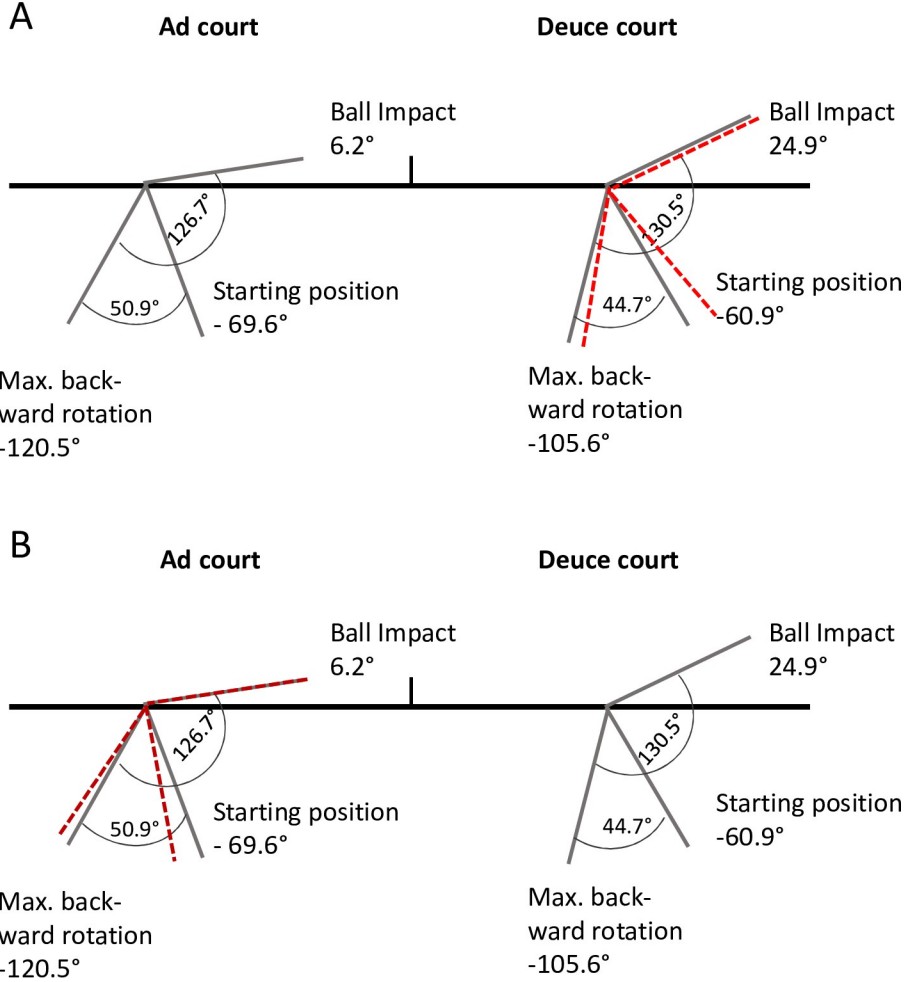

**Fig 3. Upper torso rotation at start, maximum backward rotation (clockwise), and impact on ad and deuce court side.** A) red line: mirror image transferred from the ad court kinematics to the deuce court (total ROM: 126.7˚, counter-rotation ROM: 50.9˚); B) red line: mirror image transferred from the deuce court kinematics to the ad court (total ROM: 130.5˚, counter-rotation ROM: 44.7˚).

simplify the motor learning process, a more open stance on the deuce court could be recommended. This change would adjust the total angle and body alignment towards the target, thereby leading to a more uniform and mirrored movement pattern on both service sides, as illustrated in Fig 3. This issue seems to be mainly relevant during skill acquisition in young players and might simplify the learning process in the early career stages. In the case of repeated and considerable weaknesses on one of the two service sides, even elite players could reflect a slight adjustment.

The individual service foot technique has to be considered as representing an important individual characteristic in this context. In general, there are two different types of foot techniques: the foot-up (during preparation, the back foot is moved forward next to the front foot) and the foot-back technique (players leave the back foot in relatively the same position) [38]. In our study, we noted a homogenous distribution of players using either foot technique. This is consistent with the general distribution in male elite junior players, where 52% of the players use the foot-back technique [39]. Regarding the side differences in the upper torso ROM, from a descriptive point of view, a stronger effect could be expected in the foot-back technique

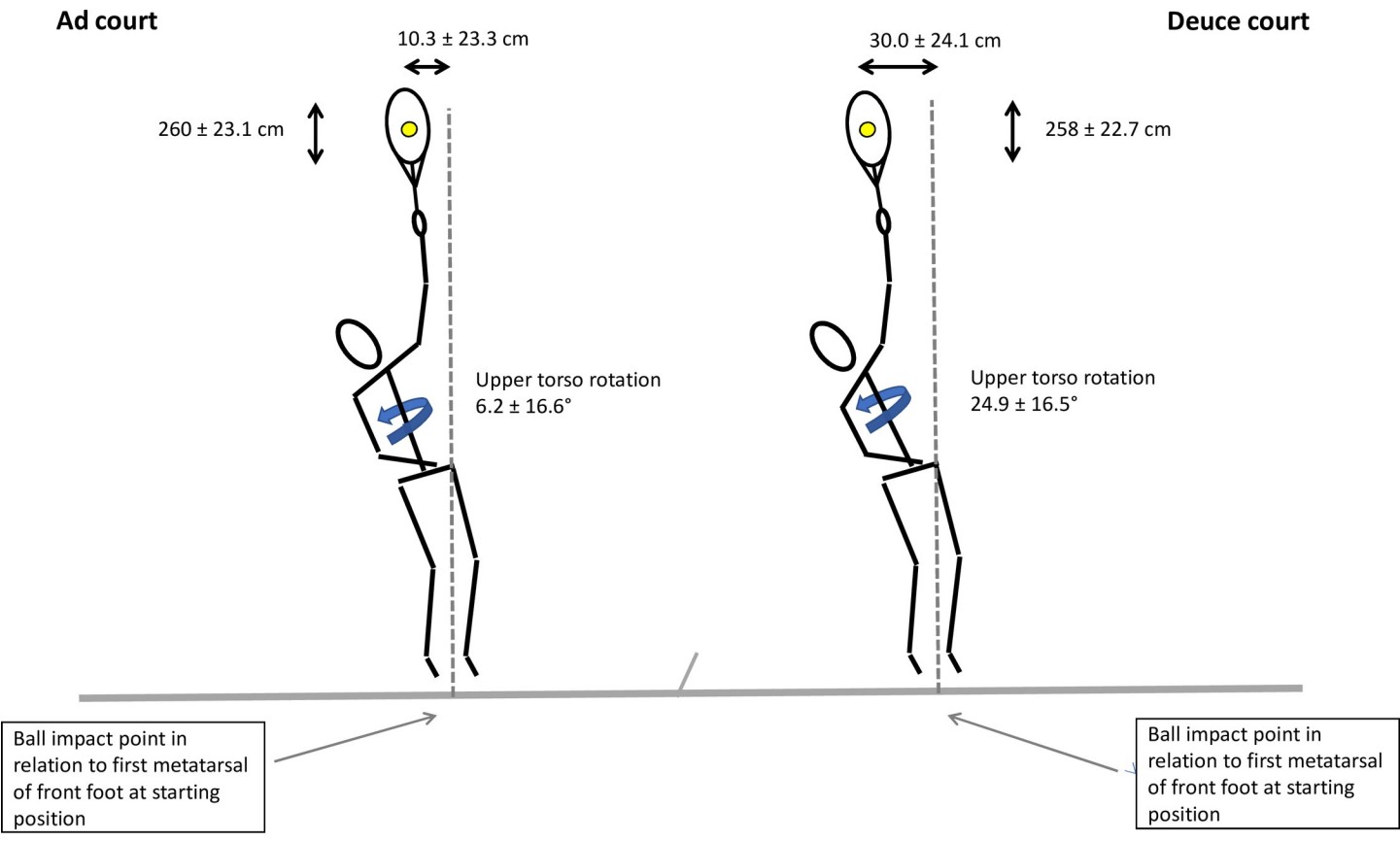

**Fig 4. Mean ball displacement and upper torso rotation at impact.**

players because the basic foot position is neutralised early during the foot-up technique (Fig 2). Nevertheless, significant interactions (side × group) are missing.

A comparison of the lateral hitting positions on both sides reveals a more leftward ball impact point on the deuce court side, with a difference of 20 cm between the two service sides (back-view position) (Table 1 and Fig 4). The lateral hitting point from the deuce court in our study (30 cm) is consistent with the findings of Reid et al. (2010) [2]. Interestingly, older professional players impact the ball less leftwards than junior players do [20, 22]. This age-dependant difference might be due to a stronger spin serve by the juniors [22], compensating for body height differences, thereby resulting in an overall lower ball impact point and, thus, the need for a more curvilinear ball trajectory above the net [20, 40]. Regarding the side differences of the ball impact points they can be explained by the side specific change in the geometric spatial relationships between the starting position and the service target. This is in line with findings of Reid et al. [22] and Carboch et al. [20], who showed that the lateral displacement at impact is significantly further left in a wide than in a T serve. This effect seems to be inevitable to avoid players being able to anticipate the service direction. When serving from different sides, by contrast, players would be able to compensate for the side-specific spatial circumstances without any tactical disadvantages as mentioned above (Fig 3).

Side specific differences of the tennis serve should also be discussed regarding possible injury-risk implications. Serve production is a violent manoeuvre generating high recurring forces and places the greatest stress on the lower back among all strokes [40–42]. Consequently, the reported high prevalence of back pain in competitive junior and professional

tennis players [23, 40, 43, 44] is not surprising, given the large loads in axial rotation [42]. The combination of repetitive rotational forces, coupled with trunk flexion and hyperextension, is particularly critical in the pathophysiology of lower back injuries [23, 42]. It is stated that players hit 50–150 serves during each of the approximately 60 matches played per season, without considering double matches and training sessions [25]. In light of our results, the higher upper torso ROM in general, compared to previous studies [18, 21], as well as the higher values on the deuce court side in comparison to the ad side (especially in foot-back technique players), could be considered as risk factors for back pain. This would underline the above-mentioned recommendation to reduce the upper torso rotation and rotational forces on the deuce court side. In this regard, it should be highlighted that we found no difference in serve velocity between the service sides, although kinematic serve differences were present. This illustrates that these side-related differences can be compensated regarding the service speed and have no effect on the power transmission to the ball. In this context, Kibler [45] stated that there are six to eight key positions during the serve, which are the most basic and which are required to be present in all motions. Further, there are multiple individual variations in other parts of the kinetic chain. It seems that small changes within the kinetic chain are compensated by subsequent body segments and their coordination.

From a practical point of view, our results clearly emphasise some modified kinematics between both service sides. Obviously, to a certain extent, the service has to be accepted as a biomechanically asymmetric and side-specific task. On the other hand, our data show some simple correction points (e. g. a change in the basic foot position) to adjust the serve kinematics which would come along several benefits. This includes a quicker and easier skill acquisition for beginners, which would allow them to generate a more stable movement execution. In tournament players, a beneficial time-saving cost–benefit ratio and a reduced injury risk can be expected.

## Limitations

Some limitations of the study design must be considered. The study took place in an indoor laboratory that did not cover the real conditions of the tennis court, which could influence the spatial orientation of the players. Further, players were fitted with retro-reflective markers placed on anatomical landmarks. In addition, the player's position along the baseline was restricted. These points could reduce the ecological validity of the investigation. Nevertheless, we used familiarisation procedures prior to the investigation in an attempt to minimise possible effects. Another limitation is that we did not perform a sample size estimation. However, the number of participants in this investigation was high compared with those of previous similar studies and the aim was to include only the best players from the respective age group. We also only analysed side-specific differences regarding serves directed to the middle of the service field (i.e. directed to the receiver' body). Further investigations are recommended to verify the changes in body kinematics between all serve directions.

## Conclusion

Changing the service side affects both the serve and the ball kinematics in elite junior tennis players. Our results underline biomechanical differences regarding the starting position (feet and upper torso), as well as movement and ball kinematics, which could be relevant for skill acquisition, injury prevention and performance enhancement. Since the serve is a highly complex and partly unsymmetrical task, the underlying reasons for these findings remain unclear. Nevertheless, a better kinematic adjustment on both sides might be an option that could economise the process of motor learning in young players. Tournament players and their coaches

are recommended to explore slight adaptations in their service positions in cases of repeated and considerable weaknesses on one of the two service sides.

## Supporting information

**S1 Dataset. Original data.**
(XLSX)

**S1 Appendix. Variables of interest.**
(PDF)

**S2 Appendix. Intra-session reliability statistics for each performance measure.**
(PDF)

**S1 Fig. Study setting.**
(PDF)

## Acknowledgments

The authors would like to thank Bruce Elliott and Jacqueline Alderson from the University of Western Australia (Perth, Australia), for assistance during data modelling to calculate relevant anatomical and ball kinematics during serves (University of Western Australia Model). Further, we would like to thank to all of the athletes for participating in the study. Also, we wish to thank Christoph Schneider for providing feedback on preparing and presenting the statistical analysis and data.

## Author Contributions

**Conceptualization:** Janina Fett, Alexander Ferrauti.

**Formal analysis:** Janina Fett.

**Funding acquisition:** Alexander Ferrauti.

**Investigation:** Janina Fett, Nils Oberschelp, Jo-Lâm Vuong.

**Methodology:** Janina Fett, Alexander Ferrauti.

**Resources:** Alexander Ferrauti.

**Writing – original draft:** Janina Fett.

**Writing – review & editing:** Janina Fett, Thimo Wiewelhove, Alexander Ferrauti.

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
