## [Decision Letter · Decision Letter 0]

3 Feb 2021

PONE-D-20-36888

Title: Kinematic characteristics of the tennis serve from the ad and deuce court service positions in elite junior players.

PLOS ONE

Dear Dr. Fett,

Thank you for submitting your manuscript to PLOS ONE. After careful consideration, we feel that it has merit but does not fully meet PLOS ONE’s publication criteria as it currently stands. Therefore, we invite you to submit a revised version of the manuscript that addresses the points raised during the review process.

In addition to the comments and suggestions of the two reviewers, I  found a serious issue, i.e., your conclusions are not supported by the data. Unless you solve  this issue, PLoS ONE cannot publish the work.

The problem is as follows. You assume, that the left and right (ad and deuce court) serve represent symmetric tasks,   but playing a tennis serve is a highly asymmetric activity, not only biomechanically but also perceptually, because the head with the eyes are aside from the arms, and because the net is not perpendicular to the ball's trajectory.

1. Given the asymmetry of the player, the orientation of the net differs significantly from the player's perspective depending on the side.

2. In a real playing situation, the player also has to be prepared for the return, which might have effects on the optimal orientation of the body. 

Second, the manuscript confuses "statistically significant results" with "biomechanically relevant factors". This is problematic, because if there are redundant degrees of freedom that do not affect the performance, the significant effects can well be biomechanically meaningless. 

In addition to the issues raised by the reviewers, these effects need to be discussed and need to be reflected in the conclusions.

We look forward to receiving your revised manuscript.

Kind regards,

Marc H.E. de Lussanet, Ph.D.

Academic Editor

PLOS ONE

Journal Requirements:

2. We noted in your submission details that a portion of your manuscript may have been presented or published elsewhere.

"Data of foot angle and upper torso rotation were presented in the book of abstract of  the  23rd  Annual  Congress  of the  European  College  of  Sport  Science.  

According to submission guidlines the prior publication of research results as a thesis, the presentation at medical or scientific conferences or the publication on preprint servers does not exclude the consideration of the manuscript."

Reviewers' comments:

Reviewer's Responses to Questions

**Comments to the Author**

1. Is the manuscript technically sound, and do the data support the conclusions?

Reviewer #1: Yes

Reviewer #2: Partly

2. Has the statistical analysis been performed appropriately and rigorously? 

Reviewer #1: Yes

Reviewer #2: Yes

3. Have the authors made all data underlying the findings in their manuscript fully available?

Reviewer #1: Yes

Reviewer #2: Yes

4. Is the manuscript presented in an intelligible fashion and written in standard English?

Reviewer #1: Yes

Reviewer #2: Yes

5. Review Comments to the Author

Reviewer #1: Comments to the Author

General comments:

This investigation aimed to analyze body and ball kinematics of flat serves from both service sides in junior tennis players. The study is well conducted and has a well-structured introduction with a thorough literature background. Procedures are described in high detail. Although as mentioned by the authors results could be affected by the sample size, data from elite junior players is interesting and adds information on the matter. I think this is a relevant article for publication in PLoS One.

Material and methods:

Subjects:

Lines 91-96: I would value more information regarding inclusion/exclusion criteria, especially concerning injury history and lower back pain. As mentioned in literature (Campbell et al., 2013, reference 22 here), players of similar characteristics as those analyzed in this study could show significant differences in some important kinematic variables depending on these issues.

Procedures:

Lines: 114-115: could this ‘center of the service field’ be considered as a flat serve to the receiver’s body? As mentioned in the discussion section this has certain importance in impact location and most likely other aspects. The whole rationale around the discussion section analyzes this specific laboratory layout. As a suggestion, since serving to different locations can significantly change kinematics, specifying that authors discuss a serve ‘to the body’ should be emphasized throughout the manuscript. Moreover, consider the possibility of including as a limitation or as future recommendation analyzing serves to other locations in other investigations.

Line 117; did the players rest between serves or only between sets?

Discussion:

Lines 246-248: I wouldn’t list differences found as surprising. You hypothesized there would be certain modifications due to changing service sides and players intuitively readjusting position in consequence.

Lines 330-332: consider clarifying this common instructional tip is not confirmed, as you say, only as long as initial position respective to the baseline is not modified in either serving side. It could be that the tip is useful as long as initial position changes.

I suggest authors include some information around possible explanations to similar serve velocities although certain kinematic differences were found between service sides. As well stated, contributors to velocity production as impact location or torso rotation may vary between positions yet serve velocity remained unaltered when comparing both sides.

Reviewer #2: - In this paper, a biomechanical analysis of two serve positions, the deuce (right, D) and the ad court (left, AD) side, is conducted. Although the paper seems to be well structured and written, and I only have several minor comments, my major concern is related to the main goal, hypothesis and practical applications.

- Since there are many confusing sentences along the paper, I would recommend a professional review of the paper.

- Although the biomechanical analysis of these two serve positions could be interesting in order to extract some practical applications regarding injury prevention programs (i.e., excessive trunk rotation, or problems in the kinetic chain), I don´t see any other key question to answer here. In fact, since there is a great variability in the serve among players, genders, and levels, it seems (at least in my point of view) to establish a general recommendation for these positions. At the end of the day, which one is more efficient under tournament conditions?. Can the authors provide information about this?

- For example, since the serve variability is clear, the authors only analyzed services targeting the center of the box. Can you please provide statistics to support this target and not others (i.e., out wide, to the receiver’s body, as you mentioned)?.

Introduction

- Line 50. Authors mentioned “skill acquisition” from beginner to elite…Can you mention some of those skills?.

- Lines 65-68. This is one of the things that confused me a lot. Where these recommendations come from?. If we take the mentioned reference, it would be non-reliable for me. What about ITF recommendations? National Tennis Federations suggestions, etc.?

- Line 68. Who assumes that?.

- Hypothesis: Can you please give some reference to support your hypothesis?.

- Hypothesis 2 is very subjective.

Materials and methods

- Ethics approval. Please check if this is the right place for it.

- Subjects. More information about the maturation status of the players would be needed (i.e., maturation offset).

- Lines 102-103. How this procedure was calculated?. It´s not easy to understand this “absorption wall” setting.

- Maybe some pics of the setting, as well as marker placement would help the reader to clearly understand the procedures.

- When the familiarization was conducted?. How many times the players repeated the experiment, and how was the reliability of the measures analyzed?

- Lines 113-114. I presume that this was a subjective feeling, right? Or was it related to the peak serve speed recorded?. If so, when?.

- About the serve protocol?. Can you please provide the average number of serves performed?. I assume that there were mistakes and not all the balls were placed exactly where the researches/players wanted. I would like to know the accuracy of the players in percentages. For example, if a player committed a lot of mistakes, a certain level of fatigue could be expected compared to other player who showed 95% of accuracy.

- Radar placement is rather high, taking into consideration that the average body height, isn´t it?.

- Maybe I´m wrong but the individual foot position and technique is not described (i.e., foot-up or foot-back?). This would definitely affect the results, right?.

- Kinematic variables: please see my previous comment. All the players followed the same foot position? (reference 15.)

Discussion

- Lines 247-251. Since the hypothesis is not really clear for me, these conclusions and suggestions are mainly speculative. In general, I found the discussion as very descriptive (repeating the results again) and speculative.

- I´m missing something in the discussion and it´s the relationship between these different techniques and the anthropometrical characteristics of the players. Did you check any relationship between modifications and for example, body height?. Shouldn´t be important in the serve?.

- Lines 261-263. This is like an impossible aim for me, since the variability in the serve is huge, and as I previously mentioned, the serve technique will depend on the players´ characteristics.

- Line 307: Ok, you mentioned for the fist time, the foot technique.

- There is no link between your kinematic data and previous studies analyzing the possible injury-risk implications (i.e., Review of tennis serve motion analysis and the biomechanics of three serve types with implications for injury. Abrams GD, et al. Sports Biomech. 2011; Upper limb joint kinetic analysis during tennis serve: Assessment of competitive level on efficiency and injury risks. Martin C, Bideau B, Ropars M, Delamarche P, Kulpa R. Scand J Med Sci Sports. 2014 Aug;24(4):700-7.). Maybe more information related to this point could be interesting for the reader.

Conclusions

- As I mentioned earlier, at the end I´m not really sure about the usefulness of these results, since the main goal of the serve (if I´m not wrong) is to generate high speed and being accurate. Regarding these factors, which were the differences between these two positions analyzed?. Moreover, are the differences reported related to an increase in the injury risk of these players?. If so, practical implications would be really interesting.

6. PLOS authors have the option to publish the peer review history of their article (what does this mean?). If published, this will include your full peer review and any attached files.

Reviewer #1: **Yes: **Joshua Colomar

Reviewer #2: No

---

## [Author Response · Author response to Decision Letter 0]

19 Apr 2021

Editor:

Dear Dr. Fett,

Thank you for submitting your manuscript to PLOS ONE. After careful consideration, we feel that it has merit but does not fully meet PLOS ONE’s publication criteria as it currently stands. Therefore, we invite you to submit a revised version of the manuscript that addresses the points raised during the review process. In addition to the comments and suggestions of the two reviewers, I found a serious issue, i.e., your conclusions are not supported by the data. Unless you solve this issue, PLoS ONE cannot publish the work.

The problem is as follows. You assume, that the left and right (ad and deuce court) serve represent symmetric tasks, but playing a tennis serve is a highly asymmetric activity, not only biomechanically but also perceptually, because the head with the eyes are aside from the arms, and because the net is not perpendicular to the ball's trajectory.

1. Given the asymmetry of the player, the orientation of the net differs significantly from the player's perspective depending on the side.

2. In a real playing situation, the player also has to be prepared for the return, which might have effects on the optimal orientation of the body. 

Second, the manuscript confuses "statistically significant results" with "biomechanically relevant factors". This is problematic, because if there are redundant degrees of freedom that do not affect the performance, the significant effects can well be biomechanically meaningless. 

In addition to the issues raised by the reviewers, these effects need to be discussed and need to be reflected in the conclusions.

We look forward to receiving your revised manuscript.

Kind regards,

Marc H.E. de Lussanet, Ph.D.

Academic Editor

PLOS ONE

Dear Dr. Marc H.E. de Lussanet,

We want to thank the editor and for their suggestions and constructive feedback. The abundance of partly contradicting stand points within international tennis experts shows that this topic should be discussed more intensively in tennis research and coaching practice. We have revised the manuscript according to the editors and reviewers’ comments and hope to answered all questions satisfactorily.

We completely agree that the tennis serve is a very complex and highly asymmetric task. This is in line with our results, showing that it is also asymmetric from a biomechanical point of view regarding the service sides. In the revised manuscript, we therefore included a more sophisticated approach of discussion including multiple reasons for our results (line 289ff). Next to methodological reasons in skill acquisition, individual characteristics, we agree that an important point can be perceptual factors. The service presents a highly asymmetric activity from a perceptual point of view. The spatial-visual orientation differs significantly between the two service sides. During preparation, players on the right court side have to turn away the head and eyes from the target field much more than on the ad court side. Thus, the visual perception is decreased on the deuce court side regarding the target field during early preparation phase. These spatial-visual determinants must be taken into account and may have an effect on the biomechanical response (Willams et al., 2000). These issues are reflected in the revised manuscript (discussion & conclusion) (line 289ff, 376).

Further, you are right, to mention that the anticipation of the receiver’s return may impact the server’s position along the baseline and the respective movement characteristics. During match play, from our point of view, this effect might be limited to changes of the server’s position along the baseline (e. g. leading to a shift of right-handed players more to the left when serving from the ad court side wide to the opponents backhand and increasing the chance to use the forehand inside-in as an effective answer to the return). In our study we therefore restricted the server´s lateral position to the maximum of 1.0m regarding the centre mark and we defined the target point in the service box (to centre of box meaning a serve to the opponent’s body). We assume that under these conditions the impact of strategical changes of the lateral serving position can be neglected. We would also defend our standpoint that under the restricted conditions of our study the trajectory of the ball is identical and symmetric in regard to the angle to the net and to the target zone as well as in regard to the height of the net. Nevertheless, we agree that these restrictions have to be discussed regarding their impact on the ecological validity of our study. Therefore, we extended this point as one major limitation of our study (line 358-362).

You also mentioned that the manuscript confuses "statistically significant results" with "biomechanically relevant factors". In this regard we cannot agree completely since the selected markers were chosen carefully according to the literature. Kibler (2014) stated that during the serve there are 244 possible degrees of freedom (DOF) in the body from the foot to the hand. Efficient mechanics in the kinetic chain can be improved by decreasing the possible DOF and most models of maximum efficiency in body motions find that limiting DOF to about six to eight maximises the total force output and minimise effort and load. These are called nodes and illustrates key positions which have been correlated with optimum force development and minimal applied loads. Of course, there are multiple variations in other parts of the kinetic chain, but these nodes and key positions are the most basic and the ones required to be present in all motions (Kibler, 2014). In our analysis we have focussed on those key positions (i.e., according to published data (Reid et al., 2014; Wagner et al., 2014; Whiteside et al., 2013)). 

In consequence we do not believe that the results are biomechanically meaningless and we also think that “biomechanical relevance” is rather difficult to define. In this regard the missing difference in serve velocity does not contradict the practical relevance of the findings since players adapted over years to their technical solution. On the other hand, we allow to address the point that the process of motor learning, skill acquisition and training of two even slightly different movements takes more time and therefore seems to be less efficient. However, in the revised version we followed your suggestion by less emphazising those findings with low effect sizes (i.e., variable: upper torso ROM = discussion was shortened). 

Journal Requirements:

PLOS ONE's style requirements have been considered.

2. We noted in your submission details that a portion of your manuscript may have been presented or published elsewhere.

"Data of foot angle and upper torso rotation were presented in the book of abstract of the 23rd Annual Congress of the European College of Sport Science. According to submission guidelines the prior publication of research results as a thesis, the presentation at medical or scientific conferences or the publication on preprint servers does not exclude the consideration of the manuscript."

Yes, you are right. A part of the presented data was presented at the ECSS Congress 2018 in Dublin and published in the conference proceedings (book of abstract), but they were not peer-reviewed. All submitted abstracts have been assessed for relevance and quality by an abstract review committee and the work was selected for oral presentation. However, the current manuscript covers a more extensive analysis and overview of the data. It presents new findings, and all contextual information (introduction, methods, findings and discussion) that were not possible to include in the mentioned conference abstract.

Reviewer 1

Reviewer #1: Comments to the Author

1) General comments:

This investigation aimed to analyze body and ball kinematics of flat serves from both service sides in junior tennis players. The study is well conducted and has a well-structured introduction with a thorough literature background. Procedures are described in high detail. Although as mentioned by the authors results could be affected by the sample size, data from elite junior players is interesting and adds information on the matter. I think this is a relevant article for publication in PLoS One.

We thank the reviewer for their time and the constructive feedback. We attempted to respond to and address all concerns raised by the reviewer. Please find our comments below.

2) Material and methods:

Subjects:

Lines 91-96: I would value more information regarding inclusion/exclusion criteria, especially concerning injury history and lower back pain. As mentioned in literature (Campbell et al., 2013, reference 22 here), players of similar characteristics as those analyzed in this study could show significant differences in some important kinematic variables depending on these issues.

Thanks a lot for this comment. We have added more detailed information regarding maturation, training details as well as regarding exclusion criteria, especially concerning the injury history (line 89ff). None of the players had any injuries or complaints. We therefore assume no disturbing impact on kinematics.

3) Procedures:

Lines: 114-115: could this ‘center of the service field’ be considered as a flat serve to the receiver’s body? As mentioned in the discussion section this has certain importance in impact location and most likely other aspects. The whole rationale around the discussion section analyzes this specific laboratory layout. As a suggestion, since serving to different locations can significantly change kinematics, specifying that authors discuss a serve ‘to the body’ should be emphasized throughout the manuscript. Moreover, consider the possibility of including as a limitation or as future recommendation analyzing serves to other locations in other investigations.

Yes, that is right. We have named it ‚centre of the service box‘ but it seems not clear at all. We have emphasized this in the introduction (research question line 78-80)) as well as in the methods section (line101-102). In addition, it is established in the limitations to recommend investigations verifying changes between all serve directions (line 368ff).

4) Line 117; did the players rest between serves or only between sets?

Between each serve there was a rest of 30 s to prepare for the next one. This information is added in the section ‘methods’ (line 120).

5) Discussion:

Lines 246-248: I wouldn’t list differences found as surprising. You hypothesized there would be certain modifications due to changing service sides and players intuitively readjusting position in consequence.

You are right, we have changed accordingly (Introduction: 76; discussion line260 ff).

6) Lines 330-332: consider clarifying this common instructional tip is not confirmed, as you say, only as long as initial position respective to the baseline is not modified in either serving side. It could be that the tip is useful as long as initial position changes.

Yes, you are right. Nevertheless, due to restructuring the discussion we deleted this point. 

7) I suggest authors include some information around possible explanations to similar serve velocities although certain kinematic differences were found between service sides. As well stated, contributors to velocity production as impact location or torso rotation may vary between positions yet serve velocity remained unaltered when comparing both sides.

We have added information regarding possible explanations for similar serve velocities although certain kinematic differences were found between service sides (line 341ff). Due to the cutback and restructuring of the discussion we have included this point, but not in detail as shown here.

The results show that the ball velocity does not change between the service sides although there are kinematic serve differences. In this regard the missing difference in serve velocity does not contradict the practical relevance of the findings since players adapted over years to their technical solution This illustrates that these side-related differences can be compensated regarding the service speed and have no effect on the power transmission to the ball. Kibler (2014) stated that during the service there are 6-8 key positions which have been correlated with optimum force development and minimal applied load and which are the most efficient methods of coordinating kinetic chain activation. Further, there are multiple individual variations in other parts of the kinetic chain (Kibler, Understanding the kinetic chain in tennis performance and injury, Aspetar Sports Medicine Journal, 2014, 3:492-497). In consequence, one possible reason could be the general high variability in the service execution so that small changes within the kinetic chain are compensated by subsequent body segments and their coordination. This reinforces the thesis that the results are primarily important for the training process and for facilitating techniqual acquisition (teaching and learning the service). 

Reviewer #2: 

1) - In this paper, a biomechanical analysis of two serve positions, the deuce (right, D) and the ad court (left, AD) side, is conducted. Although the paper seems to be well structured and written, and I only have several minor comments, my major concern is related to the main goal, hypothesis and practical applications. 

We thank the reviewer for their time and constructive feedback. We attempted to respond to and address all concerns raised by the reviewer. Please find our detailed responses and comments below.

2) - Since there are many confusing sentences along the paper, I would recommend a professional review of the paper.

The paper underwent a professional proof-reading a second time. We hope to have solved the problem.

3) - Although the biomechanical analysis of these two serve positions could be interesting in order to extract some practical applications regarding injury prevention programs (i.e., excessive trunk rotation, or problems in the kinetic chain), I don´t see any other key question to answer here. In fact, since there is a great variability in the serve among players, genders, and levels, it seems (at least in my point of view) to establish a general recommendation for these positions. At the end of the day, which one is more efficient under tournament conditions?. Can the authors provide information about this?

Nowadays, tennis experts worldwide agree that the service quality seems to be crucial and increasingly important for successful tennis. Therefore, an increasing body of literature is focused on service training including the number of serves and the optimum within-session sequence (e. g. Fernandez-Fernandez 2020). No training advices are referring to the two different service sides yet and service quality tests are usually performed only from the deuce court side (e. g. Ferrauti & Bastiaens 2007). On the other hand, the ITF rules are leading to a more or less balanced quantity of serves from both sides (slightly more from the deuce court side because all games are starting there while not all games end with a serve from the ad court side). Furthermore, experiences from the practical field are clearly showing that players develop specific service side preferences which might be attributed to biomechanical aspects. We are therefore convinced that a more sophisticated analysis on the specific demands and particularities is of practical relevance.

In general, our main goal was to determine whether there were kinematic ball and serve differences between the two service sides. Contrary to your statement and opinion, we still believe that these results are of applied interest especially with regard to the skill acquisition during motor learning. We agree that there has always been a huge variability in the serve kinematics depending on various individual characteristics (e. g. parallel baseline stands of lefthanded John McEnroe completely closed to the ad service box and completely open to the deuce box). On the other hand, from an economical point of view, you may agree that junior players will take less time to optimize two similar techniques instead of two different movements. This fact should be considered when teaching and coaching the tennis serve, especially in the learning process (skill acquisition) in beginners. The adjustments according to the service side (and not the baseline orientation) would be one possible recommendation leading to a more uniform movement pattern. Consequently, the achievement of a faster learning success would be conceivable and lead to a more economic and beneficial time-saving cost–benefit ratio.

Of course, you are right, that there are still other important key questions to answer, i.e. the connection to injury risk and prevention. This issue was added and improved in the revised manuscript. Since this connection does not apply to all parameters, we focussed on selected parameters (line 329ff). 

4) - For example, since the serve variability is clear, the authors only analyzed services targeting the center of the box. Can you please provide statistics to support this target and not others (i.e., out wide, to the receiver’s body, as you mentioned)?

The main reason was to adjust the target from both sides. The outcome would be most likely the same if we would have chosen the service box corners. But this would result in the double quantity of serves and therefore, from economic reasons, we decided to choose the centre of service box. We agree that this direction is less frequently used in professional tennis. Analyses of the 2009 French open show an approximately equal distribution of the serve directions from the deuce side (wide: 32.0%, body: 32.3%, T: 35.7%). More recent data show that in today’s tennis the directions to the wide and to the T are targeted more often (Kovalchik & Reid, J Sports Sci. Med. (2017) 16, 489-497; Krause et al, J Sports Sci. (2019) DOI: 10.1080/02640414.2019.1665245 published online). Nevertheless, the main reason for the target to the receiver’s body was a methodological one as mentioned before.

5) Introduction

- Line 50. Authors mentioned “skill acquisition” from beginner to elite…Can you mention some of those skills?.

i.e., movement execution, technical skill, coordination of kinetic chain (line 50-51)

6) - Lines 65-68. This is one of the things that confused me a lot. Where these recommendations come from?. If we take the mentioned reference, it would be non-reliable for me. What about ITF recommendations? National Tennis Federations suggestions, etc.?

The reference noted is from the online Coaching workbook from German Tennis Federation (TennisGate). These are typical recommendations especially for tennis players which are taught by coaches during skill acquisition in the early tennis training and learning process, especially for novice tennis players. Here, we have added more specific references for this, including also recommendations from the German tennis federation (Scholl (2014). Richtig Tennis Spielen – Optimales Training von Anfang an. BLV: München; Tina Hoskins (2003). The Tennis Drill Book, Human Kinetics: UK; Deutscher Tennis Bund (1981). Tennis-Lehrplan 2 Grundschläge. BLV: München). In the revised manuscript only the English written literature was added (line 69).

7) - Line 68. Who assumes that?.

Thanks, we deleted this sentence. It was too speculative.

8) - Hypothesis: Can you please give some reference to support your hypothesis?. Hypothesis 2 is very subjective.

Thanks for this comment, we changed hypothesis 1 and deleted 2. 

Since this is the first study which compares serve kinematics from the ad and the deuce service side we cannot support our hypothesis by further literature. Nevertheless, statements and feedback from the practical setting (federations and coaches) encouraged our exploration. Nevertheless, you are right to call hypothesis 2 as very subjective. We have removed this sentence to keep the result more open (line 76).

9) Materials and methods

- Ethics approval. Please check if this is the right place for it.

Ethics statement was added to the section ‘Subjects’ (line 93).

10) - Subjects. More information about the maturation status of the players would be needed (i.e., maturation offset).

Thank you for the comment. We have added more detailed information regarding maturation, i.e. peak height velocity, maturity offset (line 84-85). Further, the procedures of these parameters were added to the section ‘Anthropometric measurements’ (line 178 ff)

11) - Lines 102-103. How this procedure was calculated?. It´s not easy to understand this “absorption wall” setting. Maybe some pics of the setting, as well as marker placement would help the reader to clearly understand the procedures.

The players' task was to target the direction of receiver’s body. We transferred the dimensions of a tennis court to the laboratory setting. Triangulation allowed us to calculate the height and width of the target field, since the dimensions in the laboratory are much smaller than on the tennis court (e.g. distance of the server to the net). Of course, the target fields are only an approximation, since the trajectory of the ball was assumed to be linear and the height of the ball impact point had to be estimated beforehand based on the literature and an estimated average size of the players. 

The information on the methodical calculation was added (line 104-105).

To get a better overview we added a schematic screen of the laboratory setting (S1 Fig). 

In general, we agree that this laboratory approach can lead to a loss in ecological validity. On the other hand, it was necessary to realize a proper motion capturing without any surrounding bias.

12) - When the familiarization was conducted? How many times the players repeated the experiment, and how was the reliability of the measures analyzed?

The familiarization was conducted on the same day of testing prior to the testing protocol. The participants were fitted with all retro-reflective markers before the warm up. In consequence, they got used to it during performing the general movement preparation exercises as well as during specific activation and mobilization with elastic tubes. Also, the warm-up program consisting of submaximal serves were done in the laboratory and not on the real tennis court to get used to the specific setting. 

The players conducted the experiment once. Intra-session reliability of all variable measures was calculated. Intraclass correlation coefficient (ICC 3,1) and SEM (90% confidence limits) were assessed by the three testing trials and pooled between both service sides. Therefore, we used the spreadsheets for analysis of validity and reliability of Hopkins (Sportscience 19, 36-42, 2015 (sportsci.org/2015/ValidRely.htm). 

S2 Appendix shows reliability data for all analysed variables. Overall, ‘good’ and ‘excellent’ reliability was found for nearly each variable (ICCs ranging from 0.76 to 0.99). Front knee extension velocity and shoulder internal rotation velocity during propulsion as well as front and back knee flexion at impact showed a moderate reliability (ICCs ranging from 0.58 to 0.73) (Koo & Li, 2015).

This information was added to the section “Materials and methods” (i.e., section ‘kinematic variables’ (line 173ff) and ‘statistical analysis’ (line 194ff).

13) - Lines 113-114. I presume that this was a subjective feeling, right? Or was it related to the peak serve speed recorded?. If so, when?.

Yes, you are right. It was not related to the peak serve speed recorded, it was in accordance to their subjective feeling. This information was added (line 116).

14) - About the serve protocol?. Can you please provide the average number of serves performed?. I assume that there were mistakes and not all the balls were placed exactly where the researches/players wanted. I would like to know the accuracy of the players in percentages. For example, if a player committed a lot of mistakes, a certain level of fatigue could be expected compared to other player who showed 95% of accuracy.

All players performed 8 serves on each side. Four players failed to make three valid attempts within eight serves. There was a serve accuracy of 36.2 % (SD 9.9). Because of that, four players had to perform in total (both sides) three to four further serves to get three serves in (on each side). Instead of a total of 16 serves these players performed on average 19.5 (SD 0.58) serves at all. Regarding the total serve performed during a real match (50-150 first and second serves (Reid et al. 2008)) we think that this increase (total of 16 to 19.5 serves) will not lead to a significant fatigue compared to the other players. These information were added (line 124-126).

15) - Radar placement is rather high, taking into consideration that the average body height, isn´t it?.

Radar placement was aligned with the approximate height of ball contact (~ 3m). Ball contact in our study was around 2.60m (SD 0.23m). In consequence, we think that this is still a valid placement. We have revised the sentence in the manuscript (line 128)

16) - Maybe I´m wrong but the individual foot position and technique is not described (i.e., foot-up or foot-back?). This would definitely affect the results, right?.

- Kinematic variables: please see my previous comment. All the players followed the same foot position? (reference 15.) 

Yes, that is absolute important. That’s why we divided the sample and conducted further analysis regarding this effect (Table 2 & Figure 2).

The information about the players foot technique is described in the section ‘Subjects’ (line 86-87). In the first version it may have been insufficiently expressed. In the revision it is stated more clearly. There were six foot-up and eight foot-back technique players. Foot and upper torso position at starting position was determined before initiating the footwork. We believe that the foot technique can affect specific parameters (i.e., upper torso ROM), this is described in the section ‘discussion’ (line 304ff). From a descriptive point of view, the differences in upper torso ROM between the service sides could be affected by the foot-back technique players showing greater differences between the service sides with higher values on deuce court. 

17) Discussion

- Lines 247-251. Since the hypothesis is not really clear for me, these conclusions and suggestions are mainly speculative. In general, I found the discussion as very descriptive (repeating the results again) and speculative.

Thanks for this comment. Yes, we completely agree but we were not able to change accordingly, since there is no scientific literature on this topic yet. Therefore, our discussion has to be less literature based but rather content related following plausible argumentations. Since, it is a first impulse to this field of research the discussion remains partly speculative. Nevertheless, we tried to make the discussion more precisely with more relation to the literature. In this context, we have completely restructured the discussion section (line 251ff).

18) - I´m missing something in the discussion and it´s the relationship between these different techniques and the anthropometrical characteristics of the players. Did you check any relationship between modifications and for example, body height?. Shouldn´t be important in the serve?.

Thank you very much for the comment. Our goal was to investigate side-specified differences in ball and serve kinematics and to discuss possible reasons for this. Of course, the questions listed here are very interesting considering the influence of anthropometric factors (Bonato et al. Relationship between anthropometric or functional characteristics and maximal serve velocity in professional tennis players. J Sports Med Phys Fitness, 2014 epub ahead of print; Vaverka & Cernosek. Association between body height and serve speed in elite tennis players. Sports Biomechanics, 2013, 12:1, 30-37). However, the influence of anthropometrics was beyond the scope of our study, so that we have unfortunately not included this point in the revised manuscript. Nevertheless, it is an interesting point and we will possibly work on it later in a further re-analysis of the data.

19) - Lines 261-263. This is like an impossible aim for me, since the variability in the serve is huge, and as I previously mentioned, the serve technique will depend on the players´ characteristics.

On the one hand we completely agree but we think that our data justify at least to re-think the current tennis coaches’ textbooks, the learning methods during skill acquisition, and, in some cases, even to reflect elite player´s technique in case they show repeated considerable weaknesses on one of the two service sides. This is mentioned accordingly in the conclusions.

20) - Line 307: Ok, you mentioned for the fist time, the foot technique. 

Within the subject description, we provide information regarding the players’ foot technique. In the first version it may have been insufficiently expressed. In the new one it is stated more clearly (line 87). Because we think, that the foot technique is of great importance for the total upper torso rotation we conducted further calculation for this value (Table 2, Figure 2). These results outline that from a descriptive point of view the differences in upper torso ROM between the service sides could be affected by the foot-back technique players showing greater differences between the service sides with higher values on deuce court (line 304ff). 

21) - There is no link between your kinematic data and previous studies analyzing the possible injury-risk implications (i.e., Review of tennis serve motion analysis and the biomechanics of three serve types with implications for injury. Abrams GD, et al. Sports Biomech. 2011; Upper limb joint kinetic analysis during tennis serve: Assessment of competitive level on efficiency and injury risks. Martin C, Bideau B, Ropars M, Delamarche P, Kulpa R. Scand J Med Sci Sports. 2014 Aug;24(4):700-7.). Maybe more information related to this point could be interesting for the reader.

Thank you for this comment. You are right, implications to injury risk are missing and we have improved this. Since this connection does not apply to all parameters, we focussed on selected parameters. 

Line 329 ff:

Side specific differences of the tennis serve should also be discussed regarding possible injury-risk implications. Serve production is a violent manoeuvre generating high recurring forces and places the greatest stress on the lower back among all strokes (40–42). Consequently, the reported high prevalence of back pain in competitive junior and professional tennis players (23, 43, 44) is not surprising given the large loads in axial rotation (42). The combination of repetitive rotational forces coupled with trunk flexion and hyperextension is particularly critical in the pathophysiology of lower back injuries (23, 42). It is stated that players hit 50-150 serves during each of around 60 matches per season, not considering double matches and training sessions (25). In the light of our results the higher upper torso ROM in general compared to previous studies (18, 21) as well as higher values on the deuce court side in comparison to the ad side (especially in foot-back technique players), could be consider as a risk factor for back pain. This would underline the above-mentioned recommendation to reduce the upper torso rotation and rotational forces on the deuce court side. In this regard it should be highlighted that we found no difference in serve velocity between the service sides although there are kinematic serve differences. This illustrates that these side-related differences can be compensated regarding the service speed and have no effect on the power transmission to the ball.

22) Conclusions

- As I mentioned earlier, at the end I´m not really sure about the usefulness of these results, since the main goal of the serve (if I´m not wrong) is to generate high speed and being accurate. Regarding these factors, which were the differences between these two positions analyzed?. Moreover, are the differences reported related to an increase in the injury risk of these players?. If so, practical implications would be really interesting.

Thanks for the comment. We hope by answering the above comments, we have been able to clarify the rational behind our main goal. We agree that it is important to generate high speed and accurate precision. In this respect, there is no difference between the two service sides (see Comment 7 reviewer 1 & line 341ff). But beyond that, it can still be important for the learning process. Learning the technical skill of the service could be easier and more economic. The relationship to injury mechanics was added to the revised manuscript (line 329ff).

---

## [Decision Letter · Decision Letter 1]

20 May 2021

Title: Kinematic characteristics of the tennis serve from the ad and deuce court service positions in elite junior players.

PONE-D-20-36888R1

Dear Dr. Fett,

We’re pleased to inform you that your manuscript has been judged scientifically suitable for publication and will be formally accepted for publication once it meets all outstanding technical requirements.

Kind regards,

Marc H.E. de Lussanet, Ph.D.

Academic Editor

PLOS ONE

Additional Editor Comments (optional):

Reviewers' comments:

Reviewer's Responses to Questions

**Comments to the Author**

1. If the authors have adequately addressed your comments raised in a previous round of review and you feel that this manuscript is now acceptable for publication, you may indicate that here to bypass the “Comments to the Author” section, enter your conflict of interest statement in the “Confidential to Editor” section, and submit your "Accept" recommendation.

Reviewer #1: All comments have been addressed

Reviewer #2: All comments have been addressed

2. Is the manuscript technically sound, and do the data support the conclusions?

Reviewer #1: Yes

Reviewer #2: Partly

3. Has the statistical analysis been performed appropriately and rigorously? 

Reviewer #1: Yes

Reviewer #2: Yes

4. Have the authors made all data underlying the findings in their manuscript fully available?

Reviewer #1: Yes

Reviewer #2: Yes

5. Is the manuscript presented in an intelligible fashion and written in standard English?

Reviewer #1: Yes

Reviewer #2: Yes

6. Review Comments to the Author

Reviewer #1: (No Response)

Reviewer #2: Thanks for your comments and modifications. The paper its easier to follow and I think that it deserves to be published, as you presented some practical information regarding tennis training.

7. PLOS authors have the option to publish the peer review history of their article (what does this mean?). If published, this will include your full peer review and any attached files.

Reviewer #1: **Yes: **Joshua Colomar

Reviewer #2: No

---

## [Editor Report · Acceptance letter]

13 Jul 2021

PONE-D-20-36888R1 

Kinematic characteristics of the tennis serve from the ad and deuce court service positions in elite junior players 

Dear Dr. Fett:

I'm pleased to inform you that your manuscript has been deemed suitable for publication in PLOS ONE. Congratulations! Your manuscript is now with our production department. 

Kind regards, 

on behalf of

Dr. Marc H.E. de Lussanet 

Academic Editor

PLOS ONE